

# Genome-wide characterization and expression analysis of the JRL gene family in response to hormones and abiotic stress in tomato (*Solanum lycopersicum* L.)

Hui Li,  Hongling Hu,  Lina Cao,  Yue Zhuo,  Liting Wang,  Hui Ma and Ming Zhong

Key Laboratory of Agricultural Biotechnology of Liaoning Province, College of Biosciences and Biotechnology, Shenyang Agricultural University, Shenyang, China

## ABSTRACT

Jacalin-related lectins (JRLs), a newly discovered subfamily of plant lectins, play an important role in plant growth and development and abiotic stress response. However, in the most important model and economic plant, the tomato, little is known about this gene family. Here, we conducted a genome-wide identification and characterization of the JRL gene family in tomato. A total of eight JRL gene family members (SlJRL1–SlJRL8) were identified based on the tomato genome through bioinformatics analyses, which were unevenly distributed on four chromosomes. Chromosomal localization revealed four pairs of tandemly duplicated genes. Genome collinearity analyses showed that tomato JRL genes were more closely related to *Arabidopsis* than to rice and maize. Phylogenetic analysis showed that tomato JRL could be divided into seven subgroups, and members within each subgroup shared similar gene structures and conserved motifs. Promoter analysis revealed abundant *cis*-acting elements associated with stress-responsive and phytohormone-responsive. Finally, real-time quantitative polymerase chain reaction (qRT-PCR) was used to analyze the expression profile of *SlJRL* gene under various plant hormone and abiotic stress treatments. The results show that the SlJRL gene family exhibits tissue-specific expression patterns and responds to a variety of hormonal and stress conditions. This study systematically analyzed the genomic characteristics of tomato JRL family. Our results lay the foundation for further studies on the biological functions of tomato *JRL* genes.

## INTRODUCTION

Plant lectins are a class of proteins or glycoproteins of non-enzymatic and non-immune origin that are widely present in plants (*Claes et al., 2008*). These proteins contain one or more non-catalytic domains that reversibly bind specific carbohydrates and carbohydrate complexes. This specific binding ability determines the ability of lectins to participate in plant growth and development and to mediate responses to biotic and abiotic

Corresponding authors
Hui Ma, mahui@syau.edu.cn
Ming Zhong,
mingzhong@syau.edu.cn

stresses (*Sharon, 2007*; *Ray, Kapoor & Tyagi, 2012*; *Jung et al., 2007*; *Esch & Schaffrath, 2017*; *De Coninck & Van Damme, 2021*; *Marothia et al., 2023*).

Jacalin-related lectins (JRLs) are a recently recognized subfamily of plant lectins, which are widely distributed in plants (*Bourne et al., 2004*). Based on the structural features of their subunits, JRLs have been classified into three categories. The first class contains only one jacalin structural domain, named partial lectins. The second class not only has the jacalin domain at the N-terminal or C-terminal, but also contains other domains, and is designated as chimeric lectins. The third class contains multiple jacalin structural domains named total lectins (*Claes et al., 2008*). Although the structure of JRL proteins is very different, they all have a more stable typical jacalin structure domain. The typical jacalin domain contains 125 amino acid residues, and the conservation of the first half of the domain is usually lower than that of the second half. The N-terminal of the domain also typically contains a glycosyl recognition-associated domain that does not determine glycosyl specificity (*Raval et al., 2004*). Some conserved amino acid residues are also distributed in the typical jacalin domain, such as non-polar amino acids such as glycine and phenylalanine, which play an important role in peptide chain folding or polymerization (*Bourne et al., 2004*).

As a non-classical lectin, JRL has different characteristics from classical lectin, and the most obvious difference is that JRL can respond to abiotic stress. Studies have shown that JRL can be induced to up-regulate expression when plants are subjected to stress. The first inducible JRL protein *Orysata* was isolated and identified from rice seeds after salt stress, which had mannose-specific binding ability (*Zhang et al., 2000*). *Orysata* is not expressed in untreated plant tissues, but is rapidly expressed in roots and leaf sheaths under abiotic stress. The expression of *Lem2* (a lectin-like gene) in barley was significantly upregulated under the induction of salicylic acid (SA) and its functional analogues, and decreased under drought or abscisic acid (ABA), but did not respond to the stimulation of methyl jasmonate (MeJA) treatments (*Abebe, Skadsen & Kaeppler, 2005*). The expression level of jacalin lectin *At5g28520* was significantly increased under ABA stimulation, in response to the induction of ABA signaling pathway effect factors at seedling stage in *Arabidopsis* (*Jia & Rock, 2013*). The JRL-like protein TaJRLL1, which contains two jacalin-like lectin domains, is involved in the salicylic acid (SA)/JA signaling pathway (*Xiang et al., 2011*). *PeDJ01* from moso bamboo is up-regulated in response to salt or cold stress, highlighting its significant role in stress regulation (*Ma et al., 2021*).

In addition to the hormones treatments, JRL gene can also respond to stress treatments. For instance, the expression of *PeJRL* gene significantly upregulated under low temperature, drought, and salt stress in moso bamboo (*Zhang et al., 2022*). In wheat and *Arabidopsis thaliana*, *JRL* genes were found to have biotic and abiotic stress resistance (*Yamaji et al., 2012*; *Song et al., 2013*). Overexpression of the water hyacinth (*Eichhornia crassipes*) *JRL* gene *EcJRL-1* in *Arabidopsis Thaliana* significantly enhanced its tolerance to sulfate deficiency (*Liu et al., 2009*). Jacalin-related lectins HvHorcH participate in barley root physiological response to salt stress (*Witzel et al., 2021*). *OsJRL* expression was up-regulated under various abiotic stresses (salt, drought, high temperature and low temperature stress) in rice (*He et al., 2017*). Also, overexpression of *OsJRL40* gene enhanced rice salt tolerance

(*Gao et al., 2023*). These findings suggest that JRL motifs are important in plant response and adaptation to stress environments.

Tomato (*Solanum lycopersicum*) is a model vegetable crop cultivated worldwide, and it has significance in global agricultural production (*Liu et al., 2022*). However, it frequently suffers from undesirable environmental stresses during the growth cycle, which severely restricts the growth, development and yield of tomato. A genome-wide characterization of the JRL gene family has been performed in various plant species, and the potential significance of *JRL* family genes in plant stress resistance has been demonstrated (*Song et al., 2013*; *Zhang et al., 2022*; *Quan et al., 2023*; *Gong et al., 2024*). Genome-wide and expression-level analyses provide comprehensive insights into the complex interactions between genes and their regulatory mechanisms. Nevertheless, limited information is available regarding the evolutionary relationships or characteristics of the JRL gene family in tomato. Therefore, in the present study, we studied the physicochemical properties, chromosomal distribution, gene duplication, phylogenetic tree, gene structure, protein motifs, conserved domains, *cis*-acting elements and expression profiles of *JRL* genes in tomato. These results will provide a basis for further analysis of the role of *SlJRL* genes in plant growth and response to abiotic stresses.

## MATERIALS AND METHODS

### Plant materials and treatments

Tomato plants (*Solanum lycopersicum* Mill. cv. Ailsa Craig) were kept in a growth chamber in soil under a 25 °C/16 h in light condition, 20 °C/8 h in dark condition and 60% relative humidity. For drought and salt stress, 4-week-old plants grown in soil were watered with 20% PEG6000 (w/v) fraction and 200 mM NaCl solution and grown at normal room temperature, respectively. For cold treatment, plants were placed in a light incubator set at 4 °C for 24 h. For phytohormone treatments, tomato plants were irrigated and foliar sprayed with solution of 100 μM ABA, 50 μM MeJA and 50 μM SA, respectively (*Li et al., 2024*). The leaf samples were randomly collected after 0, 1.5, 3, 9, 12 and 24 h for different treatments, respectively. Evenly growing tomato seedlings were selected and divided into roots, stems, young leaves, flowers and mature fruits. All samples were collected for three biological replicates and each replicate consisted of ten seedlings, placed immediately in liquid nitrogen and stored in a −80 °C refrigerator for gene expression analysis.

### Identification of the *JRL* genes in tomato genomes

In this study, the complete genome was downloaded from the tomato genome database (https://solgenomics.net/) (*Fernandez-Pozo et al., 2015*). JRL protein (Pfam01419) from Pfam database of hidden markov model (HMM) spectrum, search by the HMM (https://www.ebi.ac.uk/Tools/hmmer/) to obtain the JRL potential members of the family of genes, *E*-value $\leq 1e^{-5}$ (*Wheeler & Eddy, 2013*). The hypothesized JRL protein was further confirmed by SMART (http://smart.embl-heidelberg.de) and Pfam database (http://pfam.xfam.org/). ExPASy software (https://www.expasy.org/) was used to predict the large average of the length, molecular weight (kDa), theoretical isoelectric point (pI) and hydrophilicity (GRAVY) of each tomato SlJRL protein (*Mariethoz et al., 2018*). The

WoLF PSORT program was used to generate subcellular localization of tomato SlJRLs (https://wolfpsort.hgc.jp/) (*Horton et al., 2007*). The SignalP-5.0 online server 8 was used to predict whether the proteins contained signal peptides.

## Phylogenetic and collinearity analysis of the SlJRL gene family

The full-length amino acid sequences of 46 JRL proteins from *Arabidopsis* and eight JRL proteins from tomato were analyzed by ClustalW with default settings (*Larkin et al., 2007*). MEGA (http://www.megasoftware.net/, Version 11.0) was employed to construct the phylogenetic analysis using the neighbor-joining (NJ) methods with with the maximum likelihood method (bootstrap number set to 1,000) (*Tamura, Stecher & Kumar, 2021*). The tree was edited and beautified with iTol (*Letunic & Bork, 2024*). BlastP was performed between different species and collinear relationships were searched using MCScanX, and finally visualized using the advanced circos feature in TBtools (*Krzywinski et al., 2009*; *Lavigne et al., 2008*; *Wang et al., 2024*). Non-synonymous substitution rate (Ka) and synonymous substitution rate (Ks), and Ka/Ks ratio the duplicated gene pair were calculate using KaKs_Calculator2.0 (*Wang et al., 2009*).

## Chromosome localization, gene structure and conserved motif analysis of *SlJRL* genes

The chromosomal locations of SlJRL gene family members were obtained from the Sol genome database (http://www.solgenomics.net) and mapped on their respective chromosomes using Mapchart software (*Voorrips, 2002*). TBtools were used to analyze and visualize the exon-intron structure of *SlJRL* gene (*Chen et al., 2020*). MEME software v5.0.5 (https://meme-suite.org/meme/tools/meme) was used to identify conserved motifs of SlJRLs (*Bailey et al., 2009*) and TBtools was used for visualization (*Chen et al., 2020*). The resulting *JRL* genes are renamed according to their location on the chromosome.

## Promoter *cis*-acting element analysis

The 2,000 bp promoter sequence upstream of *JRL* gene was downloaded from tomato genome and submitted to PlantCARE database (http://bioinformatics.psb.ugent.be/webtools/plantcare/html/) for screening of promoter *cis*-regulatory elements.

After statistical screening, TBtools was used to visualize possible *cis*-acting elements (*Lescot et al., 2002*).

## RNA extraction and quantitative real-time PCR (qRT-PCR) analysis

Total RNA was isolated from each sample using a plant RNA extraction kit (Tiangen, Beijing, China) and the concentration of the isolated RNA was measured by a NanoDrop 1,000 spectrophotometer (Thermo Fisher Scientific, Waltham, MA, USA). cDNA was synthesized using the SMART kit (TaKaRa, Dalian, China), two μg RNA was extracted from each sample according to the manufacturer's instructions. qRT-PCR analysis of each sample was performed in three technical replicates using a CFX96TM real-time qPCR system (Bio-Rad, USA) according to the instructions of the SYBR Green kit (Tiangen, Beijing, China). The reaction conditions were as follows: 95 °C for 2 min, 95 °C for 15 s, 60 °C for 30 s, 72 °C for 15 s, 35 cycles. The *EF1a* gene (AY905538) was used as the

internal reference (*Aoki et al., 2010*). Specific primers were designed using Primer Premier 6.0 software as described in Table S1. Three biological and three technical replicates were set for all qRT-PCR analyses. Relative gene expression was calculated using the $2^{-\triangle\triangle Ct}$ method (*Livak & Schmittgen, 2001*) and expressed as mean $\pm$ standard deviation (SD). SPSS 20.0 software was used for relative expression analysis, and Origin 9.0 software was used to complete the relative expression histogram.

## RESULTS

### Identification of the *SlJRL* genes in tomato genome

A total of eight *SlJRL* genes were identified in tomato and were renamed from *SlJRL1* to *SlJRL8* (Table 1). The CDS lengths of the *SlJRL* genes ranged from 474 (*SlJRL4*) to 1,323 bp (*SlJRL8*), and they encoded proteins ranging from 157 to 340 aa. The protein molecular weights of these proteins ranged from 17.25 kDa (*SlJRL4*) to 37.58 kDa (*SlJRL8*), with theoretical isoelectric points (pIs) between 5.17 (*SlJRL7*) and 9.16 (*SlJRL6*). All JRL proteins had a negative gravy score and were therefore hydrophilic.

SignalP-5.0 analysis showed that two SlJRL proteins contain a signal peptide, the rest of the JRL protein contains no signal peptide. Subcellular predictions showed that most tomato JRL proteins were predicted to be localized in the cytoplasm and cell wall, while some were localized in other locations, such as chloroplasts, peroxisomes, vacuoles, and mitochondria.

### Chromosomal localization, duplication and synteny analysis of the *SlJRL* genes

Chromosome localization analysis showed that eight members of the SlJRL gene family were randomly distributed on four chromosomes (Fig. 1). Each *SlJRL* gene was numbered (*SlJRL1* to *SlJRL8*) according to its physical position from the top to the bottom of the respective tomato chromosome. The largest number of SlJRL genes was found on chromosome 9 (four), with two genes on chromosome 4 and one gene on each of the remaining chromosomes. We found two tandem duplicated regions on chromosome 4 and 9, and no pairs of segmental duplicates were detected among *SlJRL* genes in tomato genome (Fig. 2, Table S2), suggesting that tandem duplication events dominated the expansion of *SlJRL* family. To further investigate the evolution of *SlJRLs*, we constructed collinearity maps of tomato with one dicotyledonous plants (*A. thaliana*) and two monocotyledonous plants (*O. sativa* and *Z. mays*). The results showed that two *SlJRL* genes were collinear with *AtJRL* genes, and none of the genes were homologous to rice and maize, indicating that the SlJRL gene family was more closely related to *Arabidopsis* than to rice and maize (Fig. S1). In this study, the Ka/Ks ratio of *SlJRL* gene and collinear *SlJRL* gene replication in tomato was calculated to investigate the SlJRL gene family and the evolution of *SlJRL* gene between species. The results showed that all four *SlJRL* genes had a Ka/Ks ratio of less than one (Table S2); thus, it is clear that the members of the SlJRL gene family underwent strong purifying selection during evolution.

**Table 1**  **The *SlJRL* family genes in tomato.**

| Gene ID | Gene name | Chromosome location | CDS length (bp) | Size (aa) | MW (KDa) | pI | GRAVY | Signal peptide | Predicted location |
|---|---|---|---|---|---|---|---|---|---|
| Solyc09g083040.1.1 | *SlJRL1* | S9: 69,134,522-69,135,028 | 507 | 168 | 18.71 | 5.37 | −0.311 | No | Cytoplasm |
| Solyc09g090445.1.1 | *SlJRL2* | S9: 70,435,162-70,437,129 | 1,130 | 284 | 30.95 | 6.61 | −0.134 | No | Chloroplast |
| Solyc04g080410.2.1 | *SlJRL3* | S4: 64,677,268-64,678,673 | 1,081 | 345 | 37.97 | 7.82 | −0.317 | No | Cytoplasm |
| Solyc09g083020.1.1 | *SlJRL4* | S9: 69,130,168-69,130,641 | 474 | 157 | 17.25 | 5.67 | −0.250 | Yes | Cell wall, Cytoplasm, Mitochondria |
| Solyc09g083030.1.1 | *SlJRL5* | S9: 69,132,760-69,133,263 | 504 | 167 | 18.52 | 7.79 | −0.311 | Yes | Cell wall |
| Solyc03g121295.1.1 | *SlJRL6* | S3: 70,929,551-70,930,980 | 838 | 202 | 22.20 | 9.16 | −0.392 | No | Cell wall, Chloroplast |
| Solyc01g006240.3.1 | *SlJRL7* | S1: 859,861-860,918 | 702 | 176 | 19.90 | 5.17 | −0.245 | No | Cell wall, Chloroplast, Cytoplasm, Peroxisome, Vacuole |
| Solyc04g080420.3.1 | *SlJRL8* | S4: 64,679,251-64,681,586 | 1,323 | 340 | 37.58 | 6.09 | −0.301 | No | Cell wall, Chloroplast, Cytoplasm |

**Notes.**

bp, base pair; aa, amino acid; MW, molecular weight; pI, isoelectric point; GRAVY, grand average of hydropathicity score.
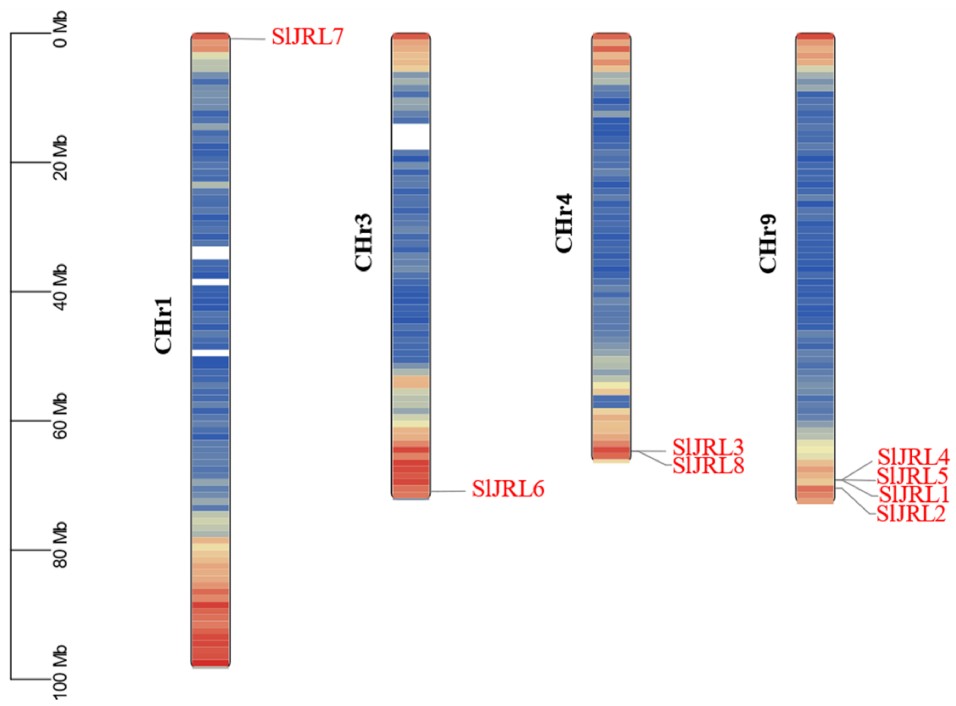

**Figure 1  Localization of *SlJRL* genes on chromosomes in tobacco.** Different colors indicate different gene densities; the redder the color, the greater the gene density; the bluer the color, the lower the gene density. The scale bar of the left displays the length of tobacco chromosomes. Different colors indicate different gene densities; the redder the color, the greater the gene density; the bluer the color, the lower the gene density. The scale bar of the left displays the length of tobacco chromosomes.

## Phylogenetic analysis of the JRL gene family

To clarify the phylogenetic relationship of *JRL* genes, we constructed a phylogenetic tree from 54 JRL protein between tomato and *Arabidopsis thaliana* (Fig. 3, Table S3). According to the protein sequences of the members of the JRL family, JRL proteins are classified into seven subgroups (I–VII). Among them, subgroup IV was the largest with 16 members, followed by subgroup V (nine), while subgroup I had the fewest members (five). The SlJRLs were distributed in subgroups V (one member), VI (three) and VII (four). Members of the JRL gene family in the same subgroup are more closely related to each other.

## Gene structure, motif, and conserved domain analysis of the tomato SlJRL gene family

To investigate the structural diversity of *SlJRL* genes in tomato, we compared the exon and intron structure of *SlJRL* gene and corresponding genomic DNA sequence by constructing an unrooted phylogenetic tree of SlJRL gene (Fig. 4A). Gene structure analysis of *SlJRL* s revealed that the number of introns ranged from zero to four among the *SlJRL* genes. *SlJRL8* has the most introns (four), while three SlJRLs completely lacked introns. The exon-intron structure patterns were commonly well-conserved in *SlJRL* s from same subgroup (Fig. 4B). We identified 10 motifs in the conserved domains of JRL proteins using the online MEME software (Fig. 4C, Table S4). Each protein contained a different number of conserved

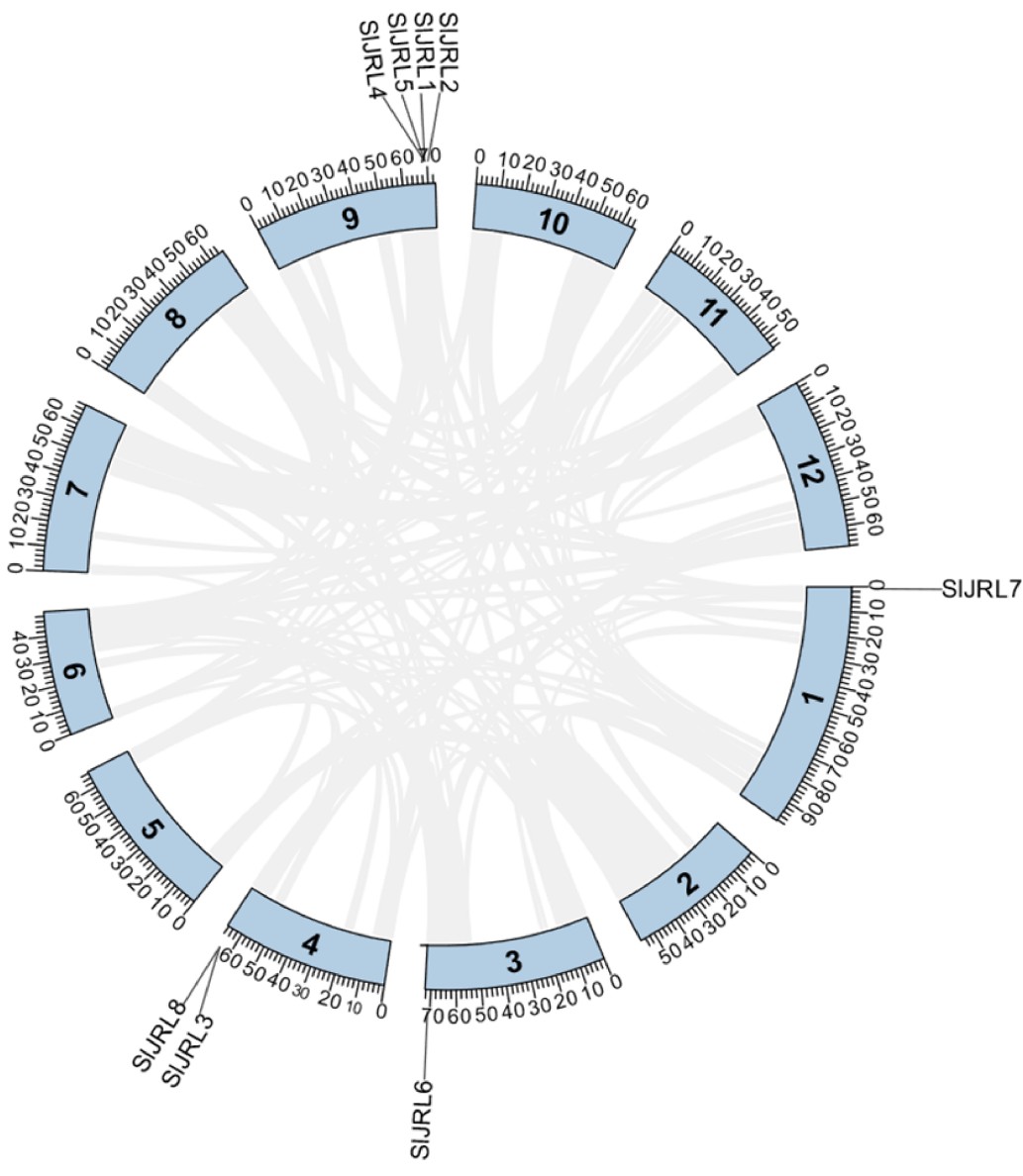

**Figure 2 Gene duplication of *SlJRL* genes on tomato chromosomes.** The tandem duplicated gene pairs are marked with red dashed lines. The corresponding relationships of duplicated gene are listed in Table S2.

motifs, ranging from four to 10, and the SlJRL members in the same subfamily also share some conserved motifs, which was consistent with the results of the phylogenetic analysis.

Using a previous classification scheme based on the number of jacalin domains and the presence or absence of other domains (*Song et al., 2013*), we identified seven type I JRL proteins and one type II JRL protein. The genes that contained only one jacalin domain had the highest percentage of type I proteins (77%). Notably, in addition to the jacalin domain, SlJRL2 also contains RXCC_like structural domains (Fig. S2).
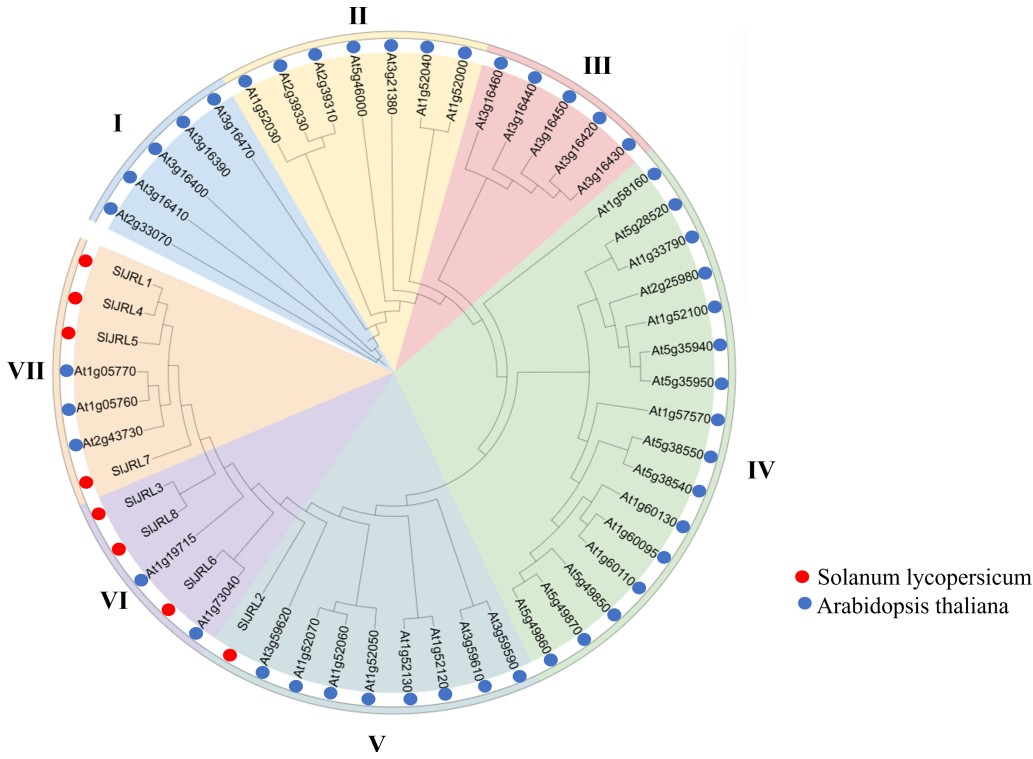

**Figure 3 Phylogenetic tree of SlJRL proteins from tomato and Arabidopsis.** The phylogenetic tree reveals distinct clades separating the SlJRL subtypes. The shading color indicates different JRL subtypes, resulting in seven subgroups of the SlJRL gene family (labeled I–VII).

## cis-element analysis of *SlJRLs*

In this study, we analyzed the 2,000 bp promoter sequence upstream of the start codon of eight *SlJRL* genes using PlantCARE (Fig. 5). The sequence analysis of eight *SlJRL* genes showed that the *cis*-acting elements of the promoter region could be divided into four categories: phytohormone responsive elements, development-related elements, abiotic and biotic stress responsive elements and light responsive elements. *SlJRL1*, *SlJRL5*, *SlJRL6*, and *SlJRL8* promoters contained *cis*-elements from all four categories. Within the hormone-responsive category, nine *cis*-regulatory elements (ABRE, TGACG-motif, TGA-element, TCA-element, AuxRR-core, TATC-box, AuxRE, P-box and O2-site) were analyzed, revealing that the ABRE motif was the most abundant. There were five *cis*-acting elements associated with development-related, such as CAT-box, AT-rich element, A-box, MRE, and circadian. There were four *cis*-regulatory elements related to stress response in *SlJRL* gene, including LTR, ARE, MBS and TC-rich repeats were identified in *SlJRL* genes, among which the ARE motif was the most common. In addition, fourteen light-responsive elements (ACE, AE-box, AT1-motif, 3-AF1 binding site, Box 4, G-box, GA-motif, GATA-motif, GT1-motif, I-box, Gap-box, TCT motif, TCCC motif and chs-CMA2a) were also analyzed, of which Box 4 was the most abundant.

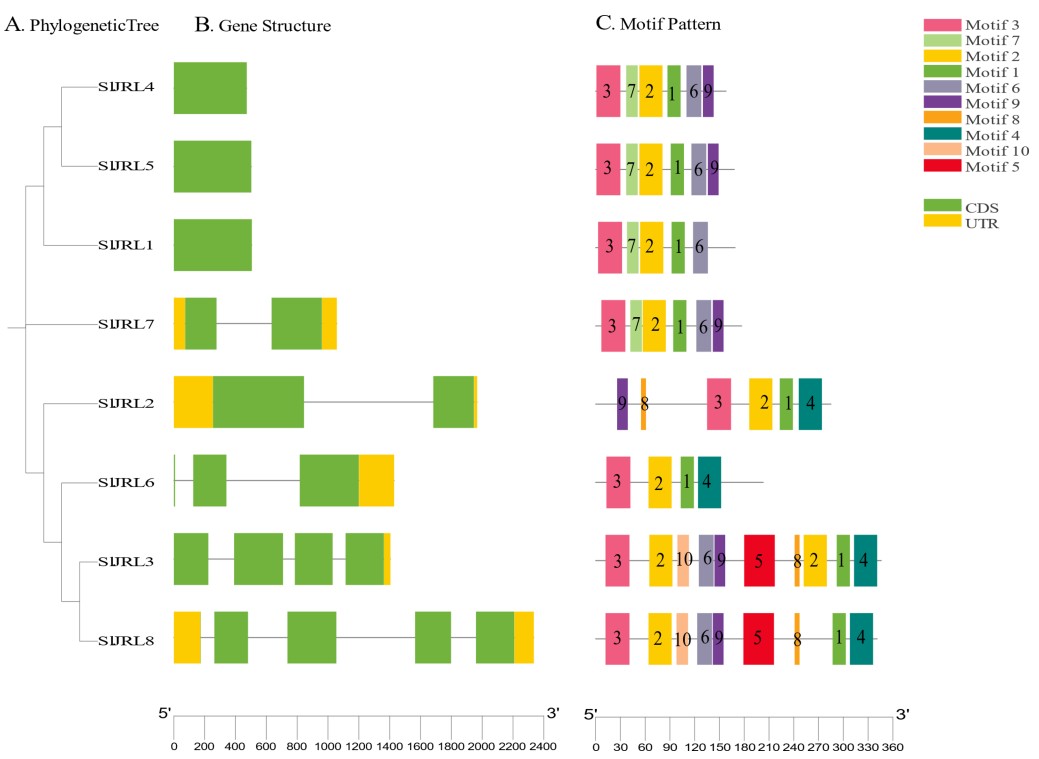

**Figure 4** **Gene structure and conserved motif analysis based on *SlJRL* phylogenetic relationships.** (A) Phylogenetic tree constructed using the NJ method with Sl JRL protein sequences. (B) Gene structure analysis of *SlJRL* genes, where blue and orange boxes represent exons and untranslated regions (UTRs), respectively, and black lines denote introns. (C) Conserved motifs in *SlJRL* genes were identified using MEME, with different colored boxes indicating distinct motifs. The scale bar of each *SlJRL* gene is shown below each gene.

## Expression pattern analysis of *SlJRL* genes

To investigate the functions of *SlJRL* genes, we used qRT-PCR method to detect the relative expression levels of eight *SlJRL* genes in different tissues such as root, stem, leaf, flower and fruit (Fig. 6). The results showed that the four tomato SlJRL genes had different expression patterns in different tissues, and *SlJRL1* and *SlJRL3* genes were not detected in all tissues. Four genes (*SlJRL2*, *SlJRL5*, *SlJRL6* and *SlJRL7*) were expressed the highest in stems, and *SlJRL8* was expressed the highest in leaves. *SlJRL8* expression was significantly higher in all tissues than the other SlJRLs, whereas *SlJRL2*, *SlJRL6* and *SlJRL7* had the lowest transcript accumulation in most tissues except the stem. *SlJRL4* had the lowest transcript accumulation in most tissues except flowers. Furthermore, *SlJRL5* were highly expressed in stems, leaves, flowers and fruits, respectively. Interestingly, except for the high expression of *SlJRL8* gene in leaf tissues, the transcription level of *SlJRL* gene in leaf tissues was lower than that in other tissues. These expression patterns may be associated with the role of the *SlJRL* genes in different tissues.

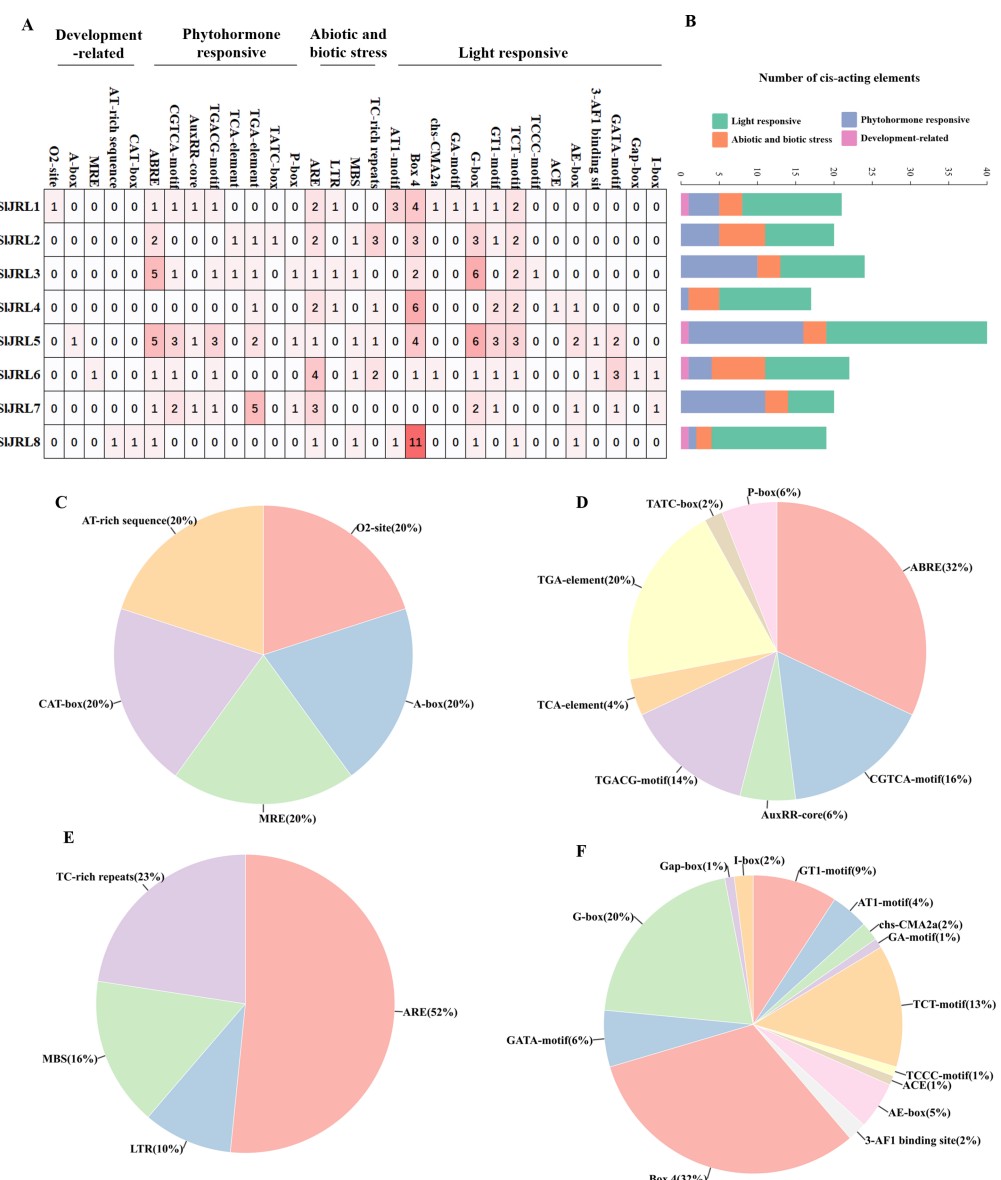

**Figure 5** *Cis*-acting elements in the promoters of *SlJRLs*. (A) The intensity of the red color and the numbers in the cells indicate the numbers of different *cis*-acting elements in each *SlJRL*. (B) The colored histograms indicate the number of different *cis*-acting elements in three categories. (C–F) The proportions of different *cis*-elements in each category: (C) phytohormone responsive, (D) development-related, (E) abiotic and biotic stress responsive and (F) light responsive.

## Expression analysis of the *SlJRL* genes under different phytohormone treatments

Promoter analysis showed that a substantial number of *cis*-acting elements associated with phytohormone responses and abiotic stress enriched in the promoter region of the *SlJRLs*, suggesting its possible involvement in these biological processes. To gain insights into the potential functions of the *SlJRL* genes in response to phytohormones,

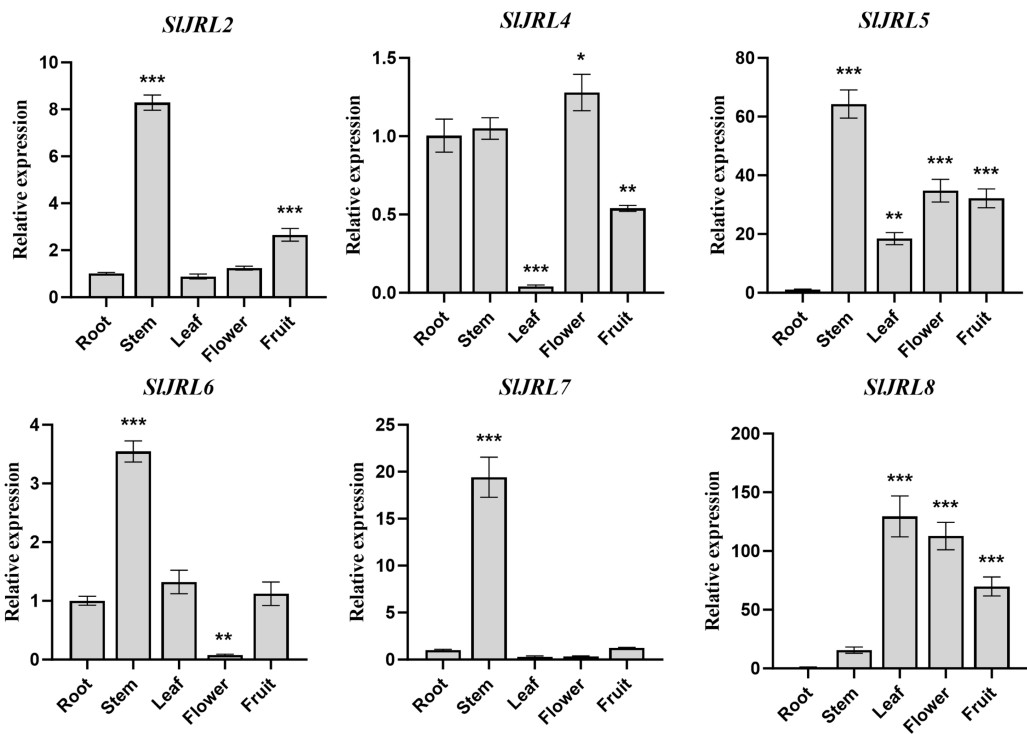

**Figure 6** **Expression analysis of *SlJRL* genes in five different tissues of tomato: root, stem, leaf, flower, mature fruit.** The standard deviations of the means of three independent biological replicates are represented by the error bars. The different asterisks above the bars indicate the significant variations between the control samples (root) and the other organs by Student's *t*-test with *p*-values less than 0.05 for *, 0.01 for **, and 0.001 for ***, respectively.

the expression patterns of *SlJRL* genes were analyzed by qRT-PCR under the treatment of three plant hormones, which were ABA, SA and MeJA (Fig. 7). After ABA treatment, the expression levels of *SlJRL2*, *SlJRL4* and *SlJRL5*, showed an up-regulated induced pattern, while *SlJRL6* and *SlJRL7* exhibited a down-regulated induced pattern, and the expression levels of *SlJRL8*, initially decreased but thereafter increased. MeJA treatment remarkably induced a up-regulated expression of *SlJRL2*, *SlJRL4* and *SlJRL6*, as well as a down-regulated expression of *SlJRL7* and *SlJRL8*. The expression levels of *SlJRL5*, initially decreased but thereafter decreased increased. Under SA treatment, *SlJRL4* and *SlJRL5* showed significantly up-regulated expression patterns, whereas *SlJRL6* and *SlJRL7* exhibited down-regulation expression levels. The expression levels of *SlJRL8*, initially decreased but thereafter increased, whereas the expression levels of *SlJRL2*, initially increased but thereafter decreased (Fig. 6B). Similarly, the expression of *SlJRL1* and *SlJRL3* were undetectable in response to the three phytohormone treatments.

## Expression analysis of the *SlJRL* genes in response to different abiotic stresses

In order to further investigate whether the expression of *SlJRL* genes was affected by abiotic stresses, we used qRT-PCR to detect the expression of eight *SlJRL* genes under

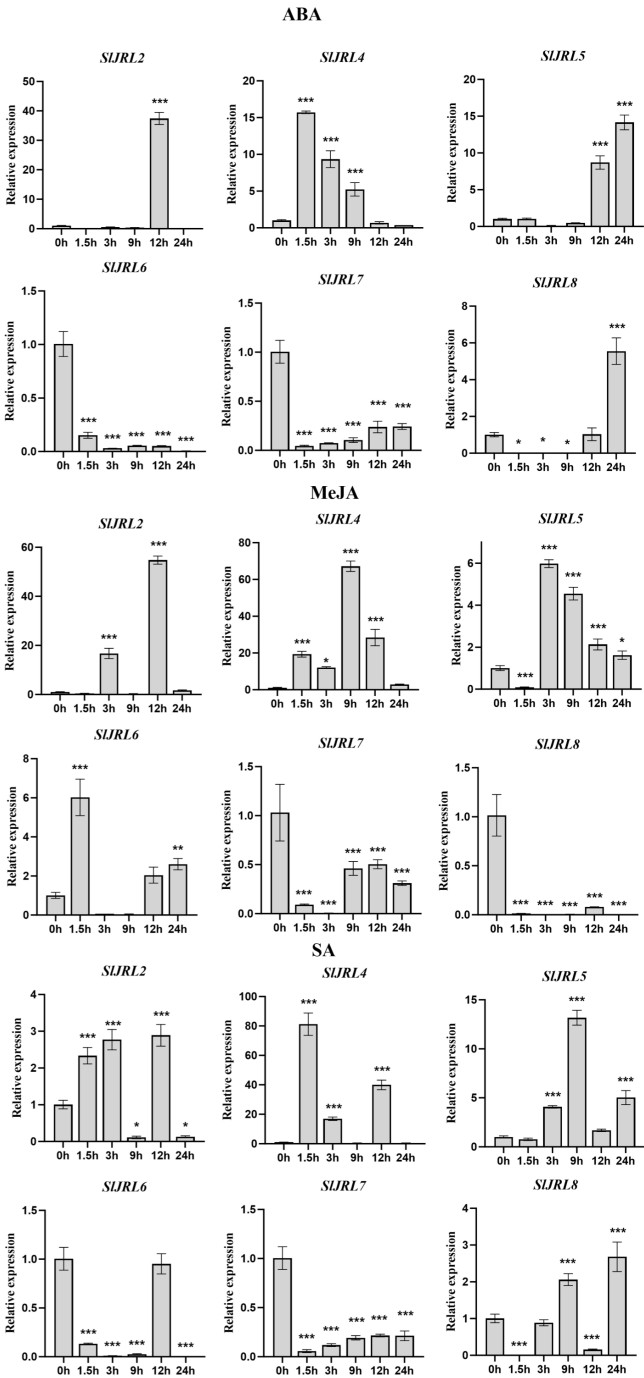

**Figure 7  Expression of tomato *SlJRL* genes under different hormones.** The standard deviations of the means of three independent biological replicates are represented by the error bars. The various asterisk marks (* for *p*-value < 0.05, ** for *p*-value < 0.01, and *** for *p*-value < 0.001) indicate statistically significant compared to its respective control using the Student's *t*-test.

salt stress (NaCl), polyethylene glycol (PEG) and low temperature conditions with 0 h untreated tomato seedlings as controls (Fig. 8). A similar situation also appeared after stress treatment, except that *SlJRL1* and *SlJRL3* were not detected, the expression of other genes was up-regulated/down-regulated induction. Among them, *SlJRL2* and *SlJRL6* were significantly upregulated, whereas *SlJRL7* showed a continuous down-regulation trend. The expression levels of *SlJRL8*, initially decreased but thereafter increased, whereas the expression levels of *SlJRL4* and *SlJRL5*, initially increased but thereafter decreased. Drought treatment significantly induced up-regulated expression of *SlJRL2*, *SlJRL4* and *SlJRL6*, and down-regulated expression of *SlJRL7* and *SlJRL8*. In addition, the expression levels of *SlJRL5*, initially increased but thereafter decreased. Low temperature treatment significantly induced up-regulated expression of *SlJRL2* and *SlJRL4*, and down-regulated expression of *SlJRL5*, *SlJRL6*, *SlJRL7* and *SlJRL8*.

## DISCUSSION

Jacalin-related lectins play an important role in plant growth, development, and abiotic stress response. Currently, the whole-genome analysis of the JRL gene family has been studied in a variety of plants, including *Arabidopsis*, *O. sativa*, wheat, *Z. mays*, sorghum, and *B. distachyon* (*Nagano et al., 2008*; *Jiang, Ma & Ramachandran, 2010*; *Song et al., 2013*; *Zhang et al., 2022*; *Quan et al., 2023*); however, little has been reported on the tomato JRL gene family. In this study, eight *SlJRL* genes were identified from the tomato genome (Fig. 1, Table 1), and the number of JRL members in tomato was much less than that in *Arabidopsis*, rice, wheat, maize and bamboo, indicating that the amplification of the SlJRL gene family is species-specific. This extension may be associated with gene duplication events (*Steven et al., 2004*). Gene amplification in plant gene families plays a crucial role in generation and amplification, which are amplified mainly by segmental and tandem replication (*Cannon et al., 2004*; *Zhu et al., 2014*; *Cao & Shi, 2012*). A total of four pairs of tandem duplication genes were identified in the SlJRL gene family of tobacco, and no obvious segmental duplication was observed (Fig. 2), indicating that tandem duplication events were the main driver of SlJRL gene family amplification. Through collinearity with other plant genome analysis, we found two homologous *JRL* gene pairs between tomato and *Arabidopsis*, 0 between tomato and rice, and 0 between tomato and maize (Fig. S1). These results indicate that the *JRL* genes of tomato are more closely related to those of *Arabidopsis* than to those of rice and maize. Phylogenetic analysis indicated that SlJRL proteins can be divided into seven groups (Fig. 3), and the *SlJRL* genes in the same group shared significant similarities in gene structure and motif composition (Fig. 4), indicating that the classification of *SlJRL* genes was relatively robust.

The study of promoter regions contributes to the understanding of gene interaction and function (*An, 1986*). *Cis*-elements play a crucial role in plant regulatory networks, contributing to a deeper understanding of transcriptional regulation and revealing the functions of the genes involved (*Hernandez-Garcia & Finer, 2014*). In this study, predictive analysis of *SlJRL* promoters revealed the presence of several elements related to growth, plant hormone response, and stress response (Fig. 5). Among them, *cis*-elements associated

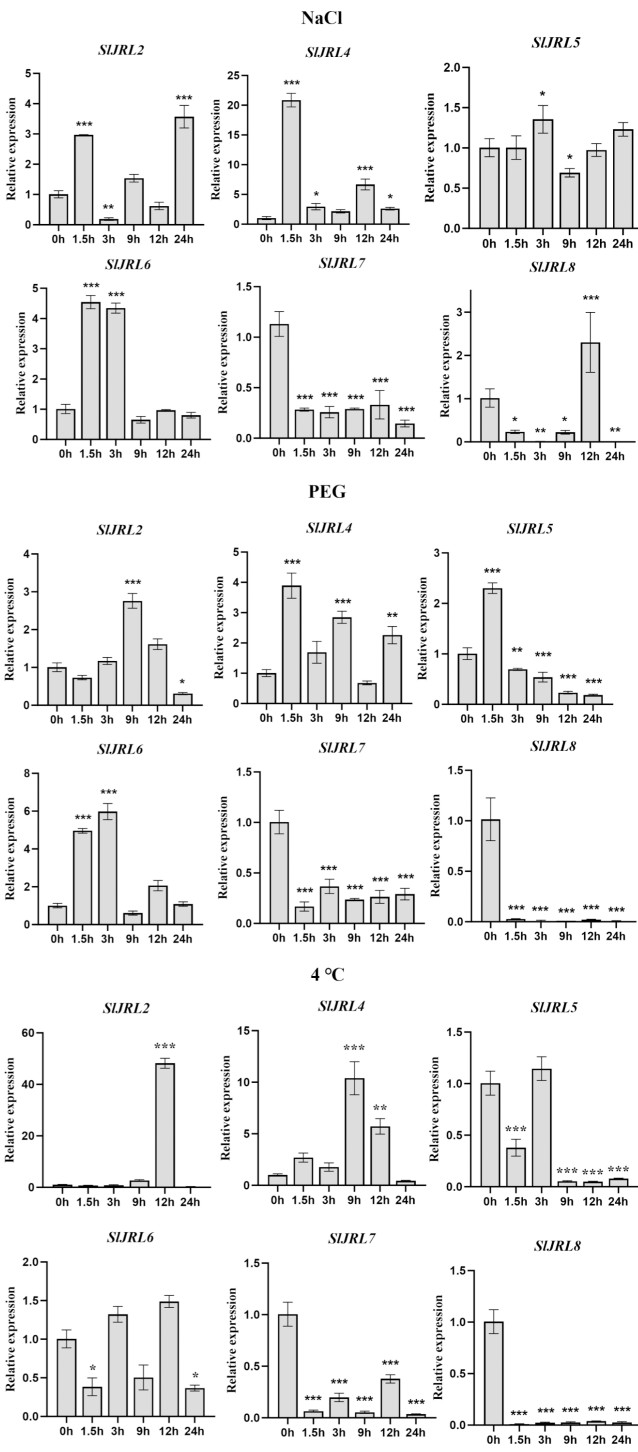

**Figure 8 Expression analysis of *SlJRL* genes under exposure to abiotic stress.** Error bars represent standard deviations of the means of three independent biological replicates. The various asterisk marks (* for *p*-value < 0.05, ** for *p*-value < 0.01, and *** for *p*-value < 0.001) indicate statistically significant compared to its respective control using the Student's *t*-test.

with hormones, such as ABA, SA, and MeJA, account for nearly 50% of *SlJRL* promoter *cis*-elements. The *SlJRL* genes contained a small number of growth response elements and a large number of stress response elements, indicating that *SlJRL* may participate in the stress responses of tomato, and may be involved in various hormonal signaling pathways. These findings suggest that the *SlJRL* genes may play a pivotal role in enhancing plant tolerance to different environmental stresses, which is also confirmed by this study.

Gene expression is associated with gene function (*Zhu et al., 2021*). The expression levels of the *SlJRL* genes were observed in different tissues. The results showed that the *SlJRL* genes were generally expressed in various organs (Fig. 6), indicating that *SlJRL* gene may play different functions in different tissues during the growth and development of tomato. *SlJRL2*, *SlJRL5*, *SlJRL6* and *SlJRL7* were highly expressed in some specific tissues, suggesting that these *SlJRL* genes in tomato have tissue specificity. Moreover, we analyzed the expression of *SlJRL* genes under abiotic stresses and hormone treatments was performed in tomato. The results showed that the expression of some *SlJRL* genes was significantly upregulated by ABA, SA, and MeJA treatments (Fig. 7), suggesting that these genes may play an important role in hormone signaling pathways. Also, we investigated and found that *SlJRL* genes respond diversely to cold, salt, and drought stresses (Fig. 8), which aligns with the reported essential roles of *JRL* genes (*Abebe, Skadsen & Kaeppler, 2005*; *Marothia et al., 2023*; *Gao et al., 2023*; *Gong et al., 2024*). Although the response mechanism of the *JRL* gene to abiotic stress remains to be further studied, these results indicate that the *JRL* gene is functionally conformed among different species. In summary, these comprehensive results laid a foundation for further research on the role of *SlJRLs* in tomato growth and development and response to environmental stress.

## CONCLUSIONS

In this study, we performed a genome-wide identification and characterization of eight *SlJRL* family members in the tomato genome, which were divided into seven groups, and distributed across four chromosomes. Chromosomal localization analysis revealed four pairs of tandem duplicated genes. *Cis*-regulatory elements responsive to environmental stress, photoresponsive, phytohormones and growth were identified in the promoters of *SlJRL* genes. The qRT-PCR results suggested that the SlJRL gene family has regulatory roles in tissue specificity and abiotic stress. These findings provide valuable information for studying the functions of *SlJRL* genes in plant growth development and responses of abiotic stress.

### Funding
This work was supported by the 2024 Liaoning Provincial Education Department Project (grant number JYTYB2024014). The funders had no role in study design, data collection and analysis, decision to publish, or preparation of the manuscript.

## Grant Disclosures

The following grant information was disclosed by the authors:
The 2024 Liaoning Provincial Education Department Project: JYTYB2024014.

## Competing Interests

The authors declare there are no competing interests.

## Author Contributions

- Hui Li conceived and designed the experiments, authored or reviewed drafts of the article, and approved the final draft.
- Hongling Hu performed the experiments, analyzed the data, prepared figures and/or tables, and approved the final draft.
- Lina Cao performed the experiments, analyzed the data, prepared figures and/or tables, and approved the final draft.
- Yue Zhuo performed the experiments, analyzed the data, prepared figures and/or tables, and approved the final draft.
- Liting Wang performed the experiments, analyzed the data, prepared figures and/or tables, and approved the final draft.
- Hui Ma analyzed the data, authored or reviewed drafts of the article, and approved the final draft.
- Ming Zhong conceived and designed the experiments, authored or reviewed drafts of the article, and approved the final draft.

## Data Availability

   The raw measurements are available in the Supplementary Files.

## Supplemental Information

Supplemental information for this article can be found online at http://dx.doi.org/10.7717/peerj.19724#supplemental-information.

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
