# Peer review of "Genome-wide characterization and expression analysis of the JRL gene family in response to hormones and abiotic stress in tomato (Solanum lycopersicum L.)"

_PeerJ, doi:10.7717/peerj.19724_

## Round 0.1 · original submission · Major Revisions

Dear Authors

The manuscript cannot be accepted for publication in its current form. It needs a major revision before publication. The authors are invited to revise the paper, considering all the suggestions made by the reviewers. Please note that the requested changes are required for publication.

With Thanks

Reviewer 1 ·

Basic reporting

No comment

Experimental design

No comment

Validity of the findings

No comment

Additional comments

This study presents a comprehensive genome-wide analysis of the Jacalin-related lectin (JRL) gene family in tomato. The study successfully identified eight SlJRL genes, mapped their chromosomal locations revealing tandem duplications, and established their evolutionary relationships with JRL genes in other plant species like Arabidopsis, rice, and maize. Furthermore, the investigation delved into the structural characteristics of these genes, their potential regulatory elements associated with stress and hormone responses, and their expression patterns under various abiotic stresses and hormone treatments, ultimately laying a valuable foundation for future functional characterization of these genes in tomato growth, development, and stress responses.
-Comments and Suggestions for Authors
- -Line 16, change “Chromosomal localization revealed four pairs of tandem duplicated genes.” To “Chromosomal localization revealed four pairs of tandemly duplicated genes.”
Line 33: Changed "non-enzymatic or non-immune origin" to "non-enzymatic and non-immune origin".
Line 36: Changed "specific binding power" to "specific binding ability"
Line 36: Also, changed "participate in the growth and development of plants and to resist biotic and abiotic stresses" to "participate in plant growth and development and to mediate responses to biotic and abiotic stresses".
Line 40: Changed "newly discovered" to "recently recognized".
Line 43: Changed "localized primarily to storage vesicles" to "primarily localized to storage vacuoles"
Line 45: Changed "JRLs were classified" to "JRLs have been classified".
Changed "preservation" to "conservation," which is the standard term in molecular biology.
Line 53: Changed "glycosyl recognition associated” to "glycosyl recognition-associated”.
Line 63: Changed "Blasticum oryzae" to the currently accepted name "Magnaporthe oryzae" and italicized it as it is a scientific name.
Changed "evidences" to "findings".
Line 92: Removed the redundant "further" before "analysis."
materials and methods section is well-written.
In Results sestion, ensure precise and consistent scientific language throughout the section
Line 175: please revise “Each SlJRL gene was named from top to bottom according to its physical position on the tomato chromosome, from 1 to 8.” To be "Each SlJRL gene was numbered (SlJRL1 to SlJRL8) according to its physical position from the top to the bottom of the respective tomato chromosome."
Line 204: "The exon-intron structure patterns were generally well-conserved..." (Removing the redundant "pattern").
Line 276: Jacalin-related lectins play an important role.
Line 284: the amplification of the SlJRL gene family is species-specific.
Lines 285-286: Gene amplification in plant gene families plays a crucial role.
Line 293: These results indicate
Line 294: Phylogenetic analysis indicated that.
Line 297: the classification of SlJRL genes was relatively robust.
Line 316-317: Revise "The results showed that some SlJRL gene in ABA, SA and MeJA treatment significantly raised..." to be "The results showed that the expression of some SlJRL genes was significantly upregulated by ABA, SA, and MeJA treatments..."
Line 318: "...may play an important role..."

Reviewer 2 ·

Basic reporting

Thank you for considering me to review the manuscript titled "Genome-wide characterization and expression analysis of the JRL gene family in response to hormones and abiotic stress in tomato (Solanum lycopersicum L.)". This manuscript presents a genome identification and functional expression analysis of the Jacalin-related lectin gene family in tomato. The study is relevant and timely, with solid methodology and data supporting the conclusions. Integrates phylogenetic, structural, and expression analyses under multiple stresses.

Suggestions
The abstract starts directly with the obtained results after the beginning two sentences, it lacks a brief description of the methodology, which should be mentioned in the Abstract.

The introduction could be improved by explaining why this genome-wide and expression-level analysis is necessary. Also, update the citations such Claes et al., 1990; Peumans and Van Damme, 1995; Peumans et al., 2000, ….

The section of materials and methods follows a standard sequence: plant materials, gene identification, phylogenetic analysis, gene structure and motif identification, promoter analysis, and gene expression. However, more details should be added on the used genotype, also the applied treatments should be subdivided and improved. The reasons for choosing the applied concentrations, time and methods should be clarified. The inconsistent terminology should be revised such as in line 99 “mass/volume fraction” should be clarified as “(w/v)”.

The section of results includes gene identification, structure, motif analysis, promoter analysis, and expression under stress/hormones. The results are supported by visual data (chromosomal mapping, phylogenetics, expression profiles). However, the expression result interpretations need more clarity. also the findings need more quantitative support when describing expression trends. Additionally, improve the figure resolution. Also, improve figure captions to be self-contained and more explanatory.

The Discussion section summarizes the obtained findings and associate them with literature. However, it suffers from redundancy and a lack of emphasis on the study novelty. Additionally, the discussion lacks a forward-looking perspective, with no suggestions for future research. To strengthen the discussion, the authors should clarify the contributions of their work, interpret the expression results more, and propose directions for further investigation.

The Conclusion section needs to be improved by highlighting the key candidate genes, briefly mentioning study constraints, and proposing next steps for applied or functional research.

The reference list suffers from inconsistencies in formatting and outdated sources. The authors should update older citations, ensure formatting consistency per journal guidelines, and use recent, species-relevant research.

Experimental design

The experimental design is appropriate

Validity of the findings

The findings are supported by robust data

---

## Round 0.2 · accepted · Accept

Dear Authors,

I am pleased to inform you that the manuscript has been improved following the last revision and can now be accepted for publication.

Congratulations on accepting your manuscript. I appreciate your interest in submitting your work to PeerJ.

With Thanks

Reviewer 1 ·

Basic reporting

no comment

Experimental design

no comment

Validity of the findings

no comment

Additional comments

The authors have made the changes I suggested in the last review. I recommend its publication in this journal.

Reviewer 2 ·

Basic reporting

The authors have addressed all previous comments, and the current version could be accepted for publication.

Experimental design

The experimental design is appropriate

Validity of the findings

The findings are supported by robust data